# CogniBias: A Benchmark for Cognitive Biases in AI–Human Dialogue

## Abstract

Cognitive biases are predictable departures from rational judgment that impact people's decisions and communication. As large language models (LLMs) are increasingly interfaced with everyday interactions, it is valuable to understand how biases arise and spread through AI–human conversations. We present CogniBias, the first benchmark dataset to study and assess cognitive biases in conversational settings. CogniBias is composed of 30+ established types of biases, like anchoring, framing, confirmation bias, and optimism bias, distilled from a variety of authentic question and answer scenarios. Each dialogue sample includes an LLM suggestion, a human-like response, and expert-informed annotations with notes, bias labels, and confidence scores. To establish the benchmark, we describe a generation pipeline incorporating multiple LLMs and include baseline results with pretrained and fine-tuned models on classification and detection tasks. Our analysis highlights the identified challenges in detecting subtle biases or overlapping biases and identified each model's frequent failures. We hope that by releasing CogniBias, it will align divergent perspectives on cognitive bias assessment and establish a baseline dataset for fairer, more trustworthy conversational AI systems.

## 1 Introduction

Cognitive biases are systematic deviations from rational judgment that shape perception, decision-making, and interaction, including well-studied cases such as anchoring, confirmation bias, framing effects, and overconfidence (6). While these insights from psychology are valuable for understanding human reasoning, they also pose risks in AI–human interaction, where conversational systems may amplify existing user biases or introduce new ones. Although recent NLP research has advanced fairness, bias detection, and debiasing, most benchmarks (e.g., WinoBias, StereoSet, CrowS-Pairs) focus on demographic or stereotypical biases (10), overlooking cognitive biases that emerge in decision-making and dialogue. Prior datasets on framing or persuasion capture isolated phenomena within narrow domains, but they lack the breadth needed to generalize across multiple categories of cognitive bias.

To address this gap, we present CogniBias, a benchmark dataset for evaluating cognitive biases in AI–human dialogue. CogniBias integrates over 30 bias types from established psychological taxonomies (e.g., anchoring, availability heuristic, sunk cost fallacy), instantiated in decision-making dialogues. Each dialogue consists of a decision-oriented question, an AI-generated suggestion, and a human-like response. Expert annotations provide bias labels and confidence scores, enabling systematic study of how biases emerge and interact in machine-mediated dialogue (8).

Our contributions are threefold:

Submitted to 39th Conference on Neural Information Processing Systems (NeurIPS 2025). Do not distribute.

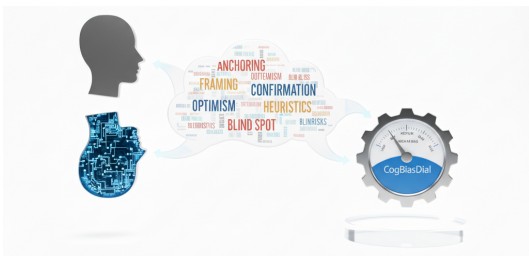

Figure 1: Motivation for CogniBias. Conversational AI may introduce or amplify cognitive biases in user decision-making. CogniBias provides the first benchmark to systematically study such effects.

- CogniBias dataset: A large-scale, multi-bias benchmark covering 30+ types of cognitive biases in AI–human dialogue.
- Generation pipeline: A reproducible LLM-based framework combining dialogue simulation with expert-informed annotations.

By releasing CogniBias, we aim to foster cross-disciplinary research at the intersection of cognitive psychology, NLP, and human–AI interaction. We envision this resource as a foundation for building conversational AI systems that are accurate, fair, and free from cognitive distortions that undermine trust and decision quality.

## 2 Related work

The NLP field has developed various benchmarks which have aimed to understand and measure bias in language models. This work has, at least to date, focused on representational-type harms related to bias based on gender, race, and/or culture. For instance, StereoSet, CrowS-Pairs and WinoBias (7; 13; 10) showed that pretrained models reinforced systemic stereotypes, and inspired new research in debiasing. However, these benchmarks and the corpus based on them, only assess demographic and social stereotypes and do not measure the cognitive processes underlying reasoning and/or decision-making.

Cognitive psychology has long documented systematic deviations in judgment, such as anchoring, framing, and the availability heuristic (5), yet these cognitive biases are largely absent from computational evaluation benchmarks. Existing NLP studies on persuasion, framing, or fallacies typically focus on one or two bias types within narrow domains like news, politics, or social media, limiting their generalizability across bias taxonomies and failing to capture conversational dynamics.

Recent research (11; 12) has indicated that large language models (LLMs) not only inherit social stereotypes, but can also exacerbate cognitive biases in dialog. For example, an anchoring effect can occur when users overweight an initial suggestion from a model, and a framing effect can bias user judgment depending on which way the information is presented. Regardless of these important considerations, there is not yet a systematic benchmark on measuring cognitive biases in a human interactive AI setting. There has been a focus on ethical risk or persuasive misuse of interacting with an LLM but no empirical datasets to use for a thorough examination of those concepts.

CogniBias differs from prior bias benchmarks by addressing cognitive rather than demographic biases within dialogue contexts 1.

## 3 Dataset design

### 3.1 Bias Taxonomy

CogniBias is founded on a classification of 30+ cognitive biases situated in cognitive psychology. These include classic heuristics and judgment errors such as anchoring, confirmation bias, framing, availability heuristic, gambler's fallacy, sunk cost fallacy, halo effect, overconfidence, optimism bias, and spotlight effect, as well as various other biases. The classification was an aggregation from the psychology literature and was modified slightly for conversational contexts to ensure that biases

Table 1: Comparative analysis of bias datasets. CogniBias uniquely focuses on cognitive biases in AI–human dialogue, unlike prior benchmarks addressing demographic stereotypes.

| Aspect | CogniBias | StereoSet | CrowS-Pairs | WinoBias |
|---|---|---|---|---|
| **Primary Focus** | Cognitive biases in AI–human dialogues (e.g., anchoring, framing, optimism bias) | Stereotypical social biases (gender, race, religion, profession) | Social biases across demographic categories | Gender bias in coreference resolution |
| **Bias Type** | 30+ **cognitive** bias types from psychology | 4 **social** bias domains | 9 **social** bias categories | Gender stereotypes only |
| **Data Format** | Three-turn dialogues: *decision question → AI suggestion → human-like response* | Triplets: *stereotype / anti-stereotype / unrelated* | Paired sentences: *stereotype vs. anti-stereotype* | Sentence templates with gendered pronouns |
| **Scale** | 3,000 dialogues; 30+ bias types; 50 simulated participants | 16,995 instances; 321 target terms | 1,508 pairs; 9 bias types | 396 examples (2 test sets) |
| **Domains Covered** | Finance, health, lifestyle, and social decision-making | Gender, race, religion, profession | Race, gender, religion, age, etc. | Occupation–gender stereotypes |
| **Interaction Type** | Conversational — includes both AI and human-like turns | Static sentence or discourse | Static sentence pairs | Static coreference sentences |
| **Uniqueness** | First dataset to study cognitive biases in AI–human dialogue | Measures demographic stereotypes only | Broad demographic coverage but not conversational | Template-based; focused narrowly on gender |
| **Human–AI Interaction** | Present — AI output may influence human bias | Absent | Absent | Absent |

covered a wide range of reasoning errors. We code each dialogue with exactly one main bias type (or No Bias, when appropriate) (6).

## 3.2 Dialogue Format

Each dataset instance follows a three-part dialogue structure:

- Question of action: A plausible situation that calls for decision-making (e.g., "Is it an opportune moment for me to invest in cryptocurrency?").
- AI proposal: The model-generated response emulates the kinds of advice, or context framing, that an LLM might use.
- Response of a human-like persona: A responding utterance replicates how a person would respond directly in a moment of cognitive bias.

Annotations include:

- Bias label (one of the 30+ categories)
- Confidence score (0–100), reflecting annotator certainty,
- Optional notes, used during quality control.

This design captures interactive dynamics where AI output may influence or reinforce human cognitive distortions.

## 3.3 Generation Pipeline

To create CogniBias, we implemented a two-stage generation pipeline:

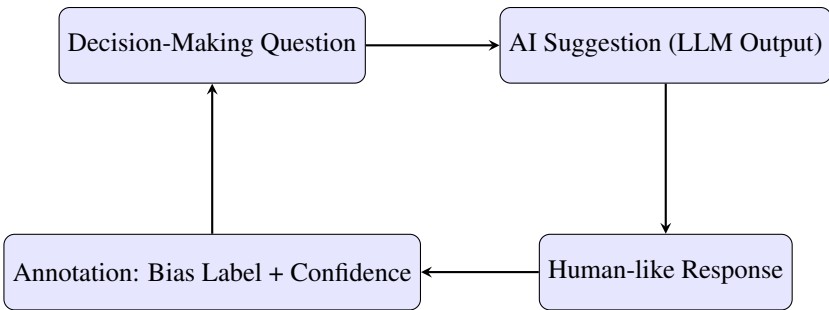

Figure 2: Overview of the CogniBias dataset generation pipeline. Each instance includes a decision-making question, an AI-generated suggestion, and a human-like response annotated with bias type and confidence score.

- Prompted LLM generation: A number of large language models (Gemini, Groq's LLaMA variants, etc.) were prompted with a pool of 60+ fallback questions covering the domains of finance, health, lifestyle, and social decision-making. The models generated both AI-generated suggestions and participant-like responses that varied stylistically and with chain of reasoning.

- Expert-informed annotation: Sampled each dialogue for one or more candidate bias from the taxonomy. Disagreement was resolved by discussion and there are now a bias label and confidence score in the final version of the annotations.

This hybrid pipeline has the benefits of leveraging the scale of LLM simulation, while using experts to maintain fidelity to psychological definitions.

## 3.4 Dataset Statistics

The released dataset includes:

- Participants: 50 simulated participants, each with a dialogue history (e.g., P01, P26)
- Dialogues: 3000 annotated dialogue samples across diverse domains.
- Bias coverage: All 30+ bias categories are represented, with frequency distributions available in Appendix A.
- Annotations: Every instance contains a bias type and confidence score (mean confidence 73 across the dataset).

## 3.5 Quality Control

Each annotation in CogniBias was validated against cognitive psychology definitions to ensure accurate bias categorization. To reduce overfitting, the dataset also includes instances labeled as No Bias. Furthermore, the questions span diverse domains such as finance, health, relationships, consumer choices, and ethics, ensuring broad applicability.

# 4 Challenges & Future Use

## 4.1 Challenges and Future Use

We highlight several open challenges for future research. Many biases are expressed subtly and implicitly, making them difficult to identify and model. Categories can overlap, as a single utterance may fit multiple bias types, and some biases occur more frequently than others, creating class imbalance. Additionally context sensitivity complicates modeling, since the same surface form may signal different biases depending on the conversational setting. Future work could explore bias-adaptive prompting strategies, where large language models dynamically adjust their communication style to mitigate bias triggers by reframing questions, providing counter-examples, or requesting

supporting evidence. Another promising direction is to treat cognitive bias detection as a multi-task, hierarchical NLP problem that captures both sentence-level and dialogue-level phenomena, accounting for overlapping biases and leveraging relationships among cognitive bias categories. By surfacing these challenges and directions, CogniBias provides not only a benchmark but also a foundation for advancing methods in bias detection, dialogue modeling, and cognitively aware AI evaluation.

# 5 Discussion & Limitations

CogniBias provides a new benchmark for evaluating cognitive biases in conversations, yet several limitations should be noted. Annotation ambiguity arises since biases often overlap (e.g., framing and optimism), and we chose to label only the dominant bias to simplify the task, which sacrifices some realism in representing the complexity of human reasoning (9). Our taxonomy covers over 30 biases but does not capture the full spectrum of distortions (e.g., moral licensing, hot-cold empathy gap) due to annotation complexity and limited examples (1). Furthermore, all dialogues are LLM-generated, which, while enabling scalability and control, reflect model-specific linguistic tendencies rather than fully naturalistic human behavior (12). The dataset also has uneven domain balance, with certain topics such as consumer decisions being overrepresented compared to others like health or finance. Annotation confidence averaged around 70, highlighting both the difficulty and subjectivity of identifying cognitive biases, which in turn increases the challenge of evaluation. Despite these limitations, CogniBias remains a systematic and scalable resource to study bias in dialogue, offering a foundation for research on bias detection, conversational fairness, and evaluation of language models grounded in psychological theory.

# 6 Broader Impact & Ethics

CogniBias is released to support research on how cognitive biases emerge in AI–human dialogue, with the aim of improving detection tools, bias-aware training, and the trustworthiness of conversational systems (11). While such a resource can advance transparency, equity, and reliability, it also carries risks of misuse, such as exploiting anchoring or framing for manipulation. To mitigate this, the dataset is framed strictly for evaluation and responsible use (12). Since the dialogues are simulated with LLMs and expert annotation, they do not fully capture cultural or contextual diversity, but future work could expand coverage across languages and settings (2). Importantly, no personal data are included, avoiding privacy concerns associated with scraped corpora (4). With these safeguards, CogniBias is intended to encourage responsible progress in conversational AI research while acknowledging both its potential and limitations. By situating biases in dialogue explicitly, it also invites interdisciplinary collaboration between AI, psychology, and ethics. This, in turn, can help inform design guidelines for safer systems. Ultimately, we hope CogniBias will serve not just as a benchmark but as a foundation for building more equitable and trustworthy AI dialogue technologies.

# 7 Conclusion

We present CogniBias, the first benchmark dataset designed for studying cognitive biases in AI–human dialogues. It includes over 30 types of biases identified in psychology and about 3,000 annotated dialogues, each consisting of a decision-oriented question, an AI recommendation, and a human-like response labeled for bias type and confidence. CogniBias enables systematic investigation of how biases manifest in conversational contexts and supports the development of bias detection or mitigation models. Our analysis highlights challenges such as overlapping categories, annotation ambiguity, and the complexity of conversational cues. By profiling cognitive biases in a structured way, CogniBias aims to inform future research across cognitive psychology, natural language processing, and human–AI interaction, fostering the development of conversational AI systems that are not only fluent and accurate but also fair, trustworthy, and aligned with human reasoning.

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

# A Bias Taxonomy

Table 2 lists the cognitive biases covered in COGNIBIAS with concise definitions.

| Bias Type | Definition |
|---|---|
| Anchoring | Relying too much on the first piece of information encountered. |
| Availability Heuristic | Judging likelihood by easily recalled examples. |
| Confirmation Bias | Seeking or interpreting information that confirms prior beliefs. |
| Framing Effect | Making different decisions depending on how information is presented. |
| Hindsight Bias | Viewing past events as more predictable than they were. |
| Loss Aversion | Preferring to avoid losses rather than acquire equivalent gains. |
| Status Quo Bias | Preferring things to stay the same instead of changing. |
| Optimism Bias | Overestimating the likelihood of positive outcomes. |
| Pessimism Bias | Overestimating the likelihood of negative outcomes. |
| Bandwagon Effect | Adopting beliefs or behaviors because others do. |
| Sunk Cost Fallacy | Continuing due to prior investments despite poor prospects. |
| Gambler's Fallacy | Believing past random events affect future probabilities. |
| Overconfidence | Overestimating one's knowledge, ability, or predictions. |
| Halo Effect | Letting one positive trait influence overall judgment. |
| Self-Serving Bias | Attributing success to oneself, failures to external factors. |
| Dunning–Kruger Effect | Overestimating competence due to limited knowledge. |
| Negativity Bias | Giving greater weight to negative experiences over positive ones. |
| Survivorship Bias | Focusing on successful cases while ignoring failures. |
| Authority Bias | Placing undue weight on the opinions of authority figures. |
| Recency Bias | Overemphasizing recent events compared to older information. |
| Outcome Bias | Judging decisions by their outcomes instead of reasoning quality. |
| Planning Fallacy | Underestimating the time or resources needed for tasks. |
| Spotlight Effect | Overestimating how much others notice one's actions or appearance. |
| Illusory Correlation | Perceiving a relationship between unrelated events. |
| Base Rate Fallacy | Ignoring statistical base rates in favor of anecdotal evidence. |

Table 2: Concise taxonomy of cognitive biases in COGNIBIAS.

# B Dataset Statistics

Table 3 reports the frequency and average annotator confidence for all 30+ cognitive biases in COGNIBIAS.

## B.1 Domain Distribution

The dataset currently covers four domains: finance, health, lifestyle, and social decision-making. In this release, domain labels are under-specified (annotated as "unknown" in the JSON files), but prompts were designed to ensure cross-domain diversity. Future expansions will include explicit domain annotations for finer-grained analysis.

# C Annotation Guidelines

This section summarizes the annotation instructions provided to expert annotators for labeling dialogues in COGNIBIAS. The goal was to ensure consistent application of cognitive bias categories across 30+ types drawn from psychology.

## C.1 General Instructions

Annotators were asked to:

- Read the dialogue in full (decision-oriented question, AI suggestion, and human-like response).
- Identify whether the human-like response exhibits a cognitive bias.

| Bias Type | Count | Avg. Confidence |
|---|---|---|
| Pessimism Bias | 121 | 60.4 |
| Optimism Bias | 110 | 82.4 |
| Framing Effect | 108 | 74.8 |
| Spotlight Effect | 108 | 66.0 |
| No Bias | 104 | 70.8 |
| Hindsight Bias | 103 | 64.4 |
| Recency Bias | 100 | 65.0 |
| Planning Fallacy | 98 | 53.8 |
| Illusion of Control | 96 | 65.3 |
| Status Quo Bias | 95 | 67.0 |
| Confirmation Bias | 95 | 87.2 |
| False Consensus Effect | 95 | 63.9 |
| Selection Bias | 94 | 64.7 |
| Pro-Innovation Bias | 93 | 65.6 |
| Outcome Bias | 92 | 65.6 |
| Just-World Hypothesis | 91 | 65.0 |
| Sunk Cost Fallacy | 91 | 65.9 |
| Dunning–Kruger Effect | 90 | 94.2 |
| Gambler's Fallacy | 89 | 64.0 |
| Halo Effect | 87 | 77.1 |
| Loss Aversion | 86 | 79.0 |
| Bandwagon Effect | 85 | 74.9 |
| Negativity Bias | 84 | 64.3 |
| Authority Bias | 84 | 82.0 |
| Actor–Observer Bias | 82 | 67.0 |
| Self-Serving Bias | 81 | 87.0 |
| Base Rate Fallacy | 81 | 63.8 |
| Overconfidence | 80 | 92.0 |
| Illusory Correlation | 80 | 65.4 |
| Anchoring | 79 | 77.7 |
| Availability Heuristic | 78 | 69.7 |
| Group Attribution Error | 71 | 65.5 |
| Survivorship Bias | 69 | 65.0 |

Table 3: Frequency and average annotator confidence for all cognitive bias categories in COGNIBIAS.

- If biased, assign exactly one **dominant bias type** from the taxonomy (see Appendix A).

- If the response was neutral, label it as **No Bias**.

- Assign a **confidence score** (0–100) indicating certainty in the label.

- Provide optional notes explaining the rationale in ambiguous cases.

## C.2 Bias Labeling Criteria

- **Dominant bias rule:** When multiple biases could apply, annotators selected the one most central to the response.

- **Surface vs. latent cues:** Explicit linguistic markers (e.g., "since it was first mentioned") were prioritized, but pragmatic and discourse-level reasoning (e.g., optimism bias without explicit markers) was also considered valid.

- **No Bias category:** Responses that were rational, balanced, or contextually neutral were marked as *No Bias*.

## C.3 Resolving Disagreements

- Each dialogue was annotated independently by two experts.

- Disagreements were discussed in weekly calibration meetings.

- In cases of persistent disagreement, the annotation was adjudicated by a third expert.

- Confidence scores were averaged across annotators for the final dataset.

These guidelines ensured that annotations were consistent, reproducible, and grounded in established cognitive psychology definitions while acknowledging the inherent subjectivity in bias identification.

## D  Dataset Format

The CogniBias dataset is released under an open license for research purposes. It is provided in JSON format with one file per participant. Each file contains a list of dialogues, where each dialogue is annotated with bias type and confidence.

### D.1  Fields

Each dialogue instance includes the following fields:

- `participant_id` – Unique identifier for the participant (e.g., P01).

- `question_id` – Identifier for the dialogue question.

- `question` – Decision-making prompt shown to the participant.

- `ai_suggestion` – Response generated by a language model.

- `human_response` – Human-like response reflecting possible bias.

- `bias_type` – Assigned label from the taxonomy (Appendix A).

- `confidence` – Annotator confidence score (0–100).

### D.2  Example JSON Snippet

Example JSON Snippet

```
{
  "participant_id": "P01",
  "dialogue": [
    {
      "question_id": "Q15",
      "question": "Should I invest in cryptocurrency right now?",
      "ai_suggestion": "It could be profitable, but it is highly
          volatile.",
      "human_response": "Yes, I ve heard so many success stories
          recently. I don t want to miss out.",
      "bias_type": "Availability Heuristic",
      "confidence": 78
    }
  ]
}
```

## E  Distribution of Bias Types

CogniBias encapsulates over 30 unique cognitive biases, amounting to 3,000 annotated dialogues collected from 50 participants. Participants showed an average confidence of 70.6 on a scale of 0–100 for each annotation. The most common biases included: Pessimism Bias, Optimism Bias, Spotlight Effect, Framing Effect, and No Bias (all present in >100 dialogues). Important, but less frequent biases included: Halo Effect, Planning Fallacy, Authority Bias, and Illusory Correlation. Figure 3 displays the frequency of the top 15 most common bias types.

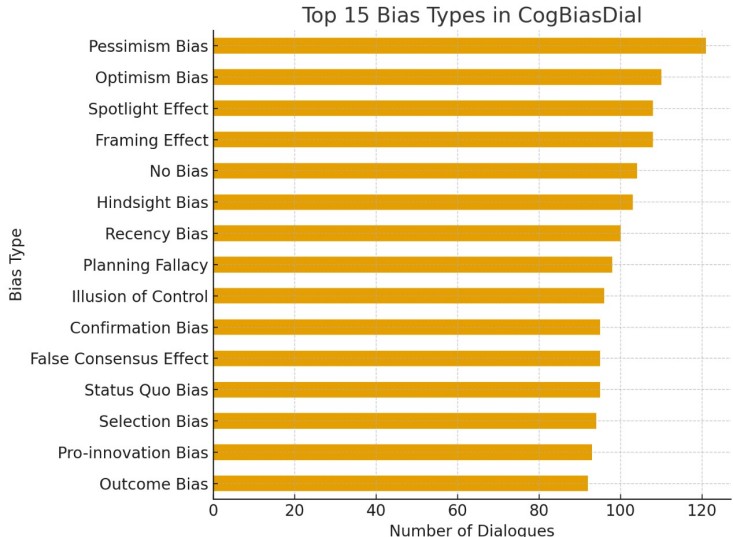

Figure 3: Distribution of the most common bias types in CogniBias. Long-tail categories ensure diversity across 30+ psychological bias types.

## F  Dataset Access

The dataset and accompanying documentation are available at: `https://github.com/Mri1306/Cognitive-Bias-Dataset`

