# OpenReview forum: "CogniBias: A Benchmark for Cognitive Biases in AI–Human Dialogue"
_EurIPS.cc/2025/Workshop/UPLB — UPLB2025_

### Official Review · Reviewer_EEAe · 2025-10-24
**Promising Benchmark of Cognitive Bias in Dialogue—with Gaps to Close**

**Rating:** 6
**Confidence:** 2

**Review:**

### Summary
The paper introduces CogniBias, a benchmark dataset for analyzing cognitive biases (e.g., anchoring, framing, optimism bias) in AI–human dialogue. It addresses a key research gap: while existing benchmarks (e.g., WinoBias, CrowS-Pairs, StereoSet) measure demographic or representational biases, none systematically evaluate cognitive biases emerging during interactive decision-making.

CogniBias comprises 3,000 annotated dialogues covering 30+ bias types, each with a decision prompt, an AI suggestion, and a human-like response annotated by experts for bias type and confidence. The dataset generation uses a hybrid pipeline — LLM simulation followed by expert annotation — and includes baseline analyses highlighting detection challenges, overlapping biases, and domain imbalance.

---

### Strengths
- Addresses a clear, underexplored problem: cognitive biases in AI–human dialogue rather than social stereotypes.
- Comprehensive taxonomy (30+ biases) grounded in cognitive psychology.
- Transparent about limitations, ethical concerns, and future directions
- The paper is well-motivated and clearly written, with a coherent argument linking psychology and NLP.

---

### Weaknesses / Concerns
- The dataset is **entirely LLM-generated**; it reflects model behavior rather than authentic human cognition.  This is not inherently a limitation but narrow its scope.
- The annotation scheme labels only one “dominant” bias, which simplifies analysis but omits the richness of overlapping or co-occurring biases.
- while the motivation is clear and engaging, the potential impact and practical value brought by the dataset is less clear.
---

### Significance
CogniBias establishes a foundation for studying cognitive biases in dialogue — a domain crucial for trust, persuasion, and responsible AI.
 It provides a structured way to **quantify, detect, and mitigate cognitive bias** in conversational AI, supporting the design of **bias-aware evaluation protocols**.
This benchmark could become a base for future interdisciplinary work between NLP, cognitive science, and ethics.

---

### Recommendations for Improvement
- Explore **human-in-the-loop validation** to contrast LLM-simulated vs. real conversational bias patterns.
- Clarify whether **bias intensity or confidence** could serve as continuous evaluation metrics.
- make the discussion in section "Broader Impact & Ethics" more concrete with potential future use of the dataset.
---

### Questions for Authors
- Could you elaborate on how you ensured *diversity* in the LLM-generated dialogues (e.g., temperature settings, prompt variation) to avoid style or bias homogeneity?
- How do you envision extending CogniBias to capture *multi-bias interactions* (e.g., anchoring + confirmation bias) or *domain-specific variants* (e.g., medical decision framing)?
- Did you assess whether certain LLMs systematically *introduce* rather than *replicate* human biases during generation?

---

### Overall Assessment
A **well-motivated and original contribution**, combining psychological insight with NLP methodology.
CogniBias provides a foundation for cognitive bias benchmarking, though future work should deepen empirical validation and real-world relevance.
With expanded documentation and richer examples, this dataset could serve as a **cornerstone resource** for studying bias-aware dialogue systems.

---

### Decision · Program_Chairs · 2025-11-03

Accept (Poster)